# *Solanum nigrum* Fruit Extract Modulates Immune System Activity of Mealworm Beetle, *Tenebrio molitor* L.

**DOI:** 10.3390/toxins15010068

**Published:** 2023-01-12

**Authors:** Arkadiusz Urbański, Natalia Konopińska, Natalia Bylewska, Radosław Gmyrek, Marta Spochacz-Santoro, Sabino Aurelio Bufo, Zbigniew Adamski

**Affiliations:** 1Department of Animal Physiology and Developmental Biology, Faculty of Biology, Adam Mickiewicz University in Poznań, 61-614 Poznań, Poland; 2Department of Sciences, University of Basilicata, 85100 Potenza, Italy; 3Laboratory of Electron and Confocal Microscopy, Faculty of Biology, Adam Mickiewicz University in Poznań, 61-614 Poznań, Poland

**Keywords:** pest control, plant extracts, insect haemocytes, humoral response

## Abstract

Here, we report the first evidence concerning the modulation of insect immune system activity after applying *Solanum nigrum* fruit extract (EXT). We focused on two main issues: (1) is EXT cytotoxic for *Tenebrio molitor* haemocytes? and (2) how EXT affects the basic immune mechanisms of *T. molitor*. The results indicate cytotoxic action of 0.01 and 0.1% EXT on beetle haemocytes. Both the injection of EXT and incubating haemocytes with the EXT solution on microscopic slides significantly increased the number of apoptotic cells. However, 24 h after injection of 0.1% EXT cytotoxic effect of the tested extract probably was masked by the increased number of circulating haemocytes. Application of 0.01 and 0.1% EXT led to impairment of the activity of basic immune mechanisms such as phenoloxidase activity and the lysozyme-like antimicrobial activity of *T. molitor* haemolymph. Moreover, the EXT elicited significant changes in the expression level of selected immune genes. However, some of the immunomodulatory effects of EXT were different in beetles with and without an activated immune system. The obtained results are an essential step toward a complete understanding of the EXT mode of action on the *T. molitor* physiology and its potential usage in pest control.

## 1. Introduction

Due to being the largest group of animals, insects severely impact on human welfare and the environment, including agricultural production. This is related to insects’ invaluable role as pollinators and the destructive effects of pest species [1]. Substantial economic losses connected to pest insects and the constantly increasing demand for food by the human population force the development of new, biosafe, and specific agents for pest control. For many years, high hopes have been associated with the use of plant extracts for this purpose [1,2].

Because of the progressive environmental changes and pesticide-resistant issue, the knowledge about potential alternatives for synthetic insecticides has significantly increased in recent years [2,3]. Natural products are already used in the markets worldwide, for example, in organic agriculture, but we are still looking for new extracts and compounds that may be used on a large scale [1]. One of the plant families that may be an excellent source of such extracts is *Solanaceae* [2,4].

*Solanum nigrum* (*Solanaceae* Juss.) is a plant native to Eurasia and introduced for example to the Americas and South Africa, which is known to produce glycoalkaloids (GAs) [4]. The research concerning the potential usage of *S. nigrum* fruit extract (EXT) clearly demonstrates that this compound can modulate many physiological events in the insect pest species, including their metabolism, development, and reproduction [4,5]. Moreover, these effects can be induced also in in vitro tests and when the EXT or GAs are administered with food [5]. However, the possible usage of EXT on a large scale seems to be the adoption of *S. nigrum* extract as a supportive addition for insecticide action. Spochacz et al. [6] demonstrated that sub-lethal doses of EXT can enhance the activity of other insecticides, such as fenitrothion [6]. The undoubted advantage of EXT is the fact that its composition was characterized, for example by Chowański et al. [7] and Yang et al. [8]. The EXT consists of more than 150 compounds, including GAs (mainly solasonine and solamargine), flavonoids, steroids saponins, amino acids and caffeoylquinic acids. However, the biological actions of EXT were mainly linked to the presence of GAs and flavonoids [7,8,9].

To the best of our knowledge, the complex effect of EXT on the insect immune system activity has never been tested. For these reasons, the initial step in our research was the evaluation of the impact of EXT on the primary immune mechanisms of the mealworm beetle *Tenebrio molitor*.

The insect immune system activity is based on the innate mechanisms, which are usually classified to cellular or humoral responses [10]. The cellular response refers to immune mechanisms associated with the haemocytes’ activity, such as phagocytosis, nodulation, or encapsulation [11,12]. The humoral response is linked to the activity of immune-related molecules, for example, antimicrobial peptides (AMPs), phenoloxidase, or lysozyme [13,14]. However, the classification of cellular and humoral response seems to be artificial because mechanisms classified to these two types of immune responses are dependent on each other [15]. 

In our study, we focused on two main issues: (1) is the EXT cytotoxic for *T. molitor* haemocytes? and (2) how the EXT affects immune mechanisms such as phenoloxidase (PO) activity or expression level of immune-related genes? Due to current literature data concerning insect immune system functioning and its response to stress conditions, the basic activity of immune mechanisms was tested, and the effects of EXT on beetles with an activated immune system were analysed.

The knowledge about the nature of the effects elicited by plant-derived compounds can be helpful in planning strategies for pest control [2,5,6]. This is related to the fact that even sub-lethal concentrations may cause insect immune system activity malfunctions and synergistically support the action of other compounds used in pest control [4,5,6]. Moreover, the used insect species are not accidental. The mealworm beetle, *T. molitor* is very often used as a model organism in toxicological, physiological, and neuro-endocrinological studies, as well as in studies concerning insect immune functioning [16]. Moreover, this species is a serious pest of stored crops. Yearly, *T. molitor* causes losses of up to 15% of the worldwide production of grains and flour [2].

## 2. Results

### 2.1. Apoptotic Ratio

The EXT induced apoptosis in haemocytes of *T. molitor* (Figure 1 and Figure 2). This cytotoxic effect was observed after EXT injection and in in vitro treatment when the tested extract was only added to the physiological saline during the incubation of haemocytes on microscopic slides. However, a significant increase in the number of apoptotic haemocytes after EXT injection at a concentration of 0.01% was only observed (Mann Whitney *U* test; U = 194.00, *p* ≤ 0.01). In the case of the direct effect of EXT on haemocytes, both tested concentrations significantly enhanced the number of apoptotic cells (Mann Whitney *U* test; 0.01% EXT − U = 200.00, *p* ≤ 0.01; 0.1% EXT − U = 289.50, *p* ≤ 0.05).

### 2.2. Total Haemocyte Count

The number of circulating haemocytes significantly increased after the injection of EXT (Figure 3). However, despite the same tendencies after application of EXT at concentrations of 0.01% and 0.1%, the statistically significant increase in THC value only at the higher of the two used concentrations was observed (*t*-test with Welch’s correction, *t* = 2.46; *p* ≤ 0.05).

### 2.3. Phenoloxidase Activity

A significant effect of EXT on *T. molitor* immune system activity was also observed in the case of PO activity (Figure 4). Injection of EXT at concentrations of 0.01% and 0.1% led to a statistically significant decrease in the PO activity (*t*-test with Welch’s correction, 0.01% EXT - *t* = 2.57; *p* ≤ 0.05; 0.1% EXT - *t* = 2.19; *p* ≤ 0.05). Compared to the control individuals the enzyme activity was lower for −22.55 ± 6.55% (0.01% EXT) and −16.92 ± 6.13% (0.1% EXT) (mean ± SEM). Moreover, similar significant changes in the PO activity, and also in immunised beetles, were observed (*t*-test with Welch’s correction, 0.01% EXT - *t* = 2.37; *p* ≤ 0.05; 0.1% EXT - *t* = 7.96; *p* ≤ 0.0001). However, compared to beetles without an activated immune system, EXT, at a concentration of 0.01%, elicited a smaller effect on PO activity (reduction of −6.97 ± 4.62%; mean ± SEM).

### 2.4. Lysozyme-like Antimicrobial Activity of T. molitor Haemolymph

Similar to PO activity, the injection of EXT caused a decrease in the lysozyme-like antimicrobial activity of *T. molitor* haemolymph (Figure 5). However, this immunostatic action of EXT only in beetles with an activated immune system was observed. In this group, both EXT concentrations significantly disrupted the lysozyme-like activity of haemolymph (Mann Whitney *U* test; 0.01% EXT − U = 92.00, *p* ≤ 0.05; 0.1% EXT − U = 90.00, *p* ≤ 0.05).

### 2.5. Expression of Immune-Related Genes

The RT-qPCR assay results partially supported the PO activity and lysozyme-like activity of *T. molitor* haemolymph. Interestingly, the expression level of immune-related genes changed differently in beetles with and without an activated immune system (Figure 6, Figure 7 and Figure 8).

#### 2.5.1. Cecropin

Twenty-four hours after injection of EXT at concentrations of 0.01% and 0.1%, the gene encoding cecropin expression level did not change. However, in beetles with an activated immune system, the expression level of the gene for cecropin significantly decreased after 0.1% EXT treatment (Mann Whitney *U* test; 0.01% EXT − U = 11.00, *p* ≤ 0.05) (Figure 6).

#### 2.5.2. Tenecin 3

Injection of 0.01% EXT significantly enhanced the level of expression of gene encoding tenecin 3 (Mann Whitney *U* test; U = 7.00, *p* ≤ 0.01) (Figure 7). Among control individuals and beetles treated with 0.1% EXT, significant differences were not observed. On the other hand, in the case of beetles with the activated immune system, applying EXT at both tested concentrations did not change the gene expression level for tenecin 3.

#### 2.5.3. Toll Receptor

Similar to the expression level of the gene encoding tenecin 3, injection of 0.01% EXT caused significant overexpression of the gene for Toll receptor (Mann Whitney *U* test; U = 10.00, *p* ≤ 0.01) (Figure 8). This effect was not observed in the case of beetles treated with EXT at concentrations of 0.1%. The injection of EXT and then activation of *T. molitor* immune system led to significant downregulation of gene encoding Toll receptor (0.01% EXT—Mann Whitney *U* test; U = 17.00, *p* ≤ 0.05; 0.1% EXT—*t*-test with Welch’s correction, t = 2.22, *p* ≤ 0.05) (Figure 8).

## 3. Discussion

Here, we report the first evidence concerning the impairment of immune system activity of insects after the application of *S. nigrum* fruit extract, which is a crucial step in fully understanding the mode of action of EXT on *T. molitor* physiology and its potential usage in pest control.

The influence of EXT on insect haemocytes indicates the strong cytotoxic effect of the tested extract, even at sub-lethal concentrations. This was especially visible when EXT was added directly to the suspension of haemocytes incubated on the microscopic slide. This kind of treatment caused a significant increase in the number of apoptotic haemocytes. Interestingly, 24 h after injection of EXT, the cytotoxic effect was only revealed at the lower concentration of 0.01%. The lack of statistically significant results after the injection of 0.1% EXT is probably closely related to the results of THC. The determination of the number of circulating haemocytes showed that the 0.1% EXT led to an increase in THC value. These results suggest that the potential cytotoxic effect after the injection of 0.1% EXT could be masked by releasing a new pool of haemocytes from haemocytopoietic organs. A similar effect was observed by Harikrishnan et al. [17] in the research on tiger shrimp *Penaeus monodon*. Supplementation of EXT with food caused, for example, an increase in the THC value and improved the general health status of tested crustacean species [15]. We did not exclude that the observed increase in THC value after application of EXT may also be related to hormonal regulation of stress response. Research by Adamo [18] or Urbański et al. [19] may suggest that hormones participating in this process, such as octopamine and tachykinin-related peptides, increase the number of haemocytes. However, further research is needed to explain this phenomenon. 

The results obtained after EXT treatment should be also linked to the direct action of the main EXT components, especially GAs—solanine and solamargine [5,6,7]. Recent literature data demonstrated that solanine possesses pro-apoptotic properties against different cell populations, such as human pancreatic cancer cells or human trophoblast cells [20,21]. Pro-apoptotic properties of solamargine were also proven. For example, this compound induces apoptosis in human hepatoma cells [22]. In addition, not only GAs may promote apoptosis, but also other EXT components [23,24,25]. Research conducted by Huang, Syu and Lin [9] demonstrated that extract from mixed parts of *S. nigrum* (including fruits) can contain many polyphenolic compounds, which also possess proapoptotic activity [23,24,25]. For example, research by Wang, Chung, Wu, Lan, Yang and Wang [23] demonstrated that the *S. nigrum* polyphenolic extract inhibits growth of hepatocarcinoma cells, among others, by induction of caspases-3, 8, and 9 cleavages. Moreover, the gentisic acid, the most abundant polyphenols in extract tested by Huang, Syu and Lin [9], itself can induce apoptosis. Research by Dong et al. [26] demonstrated that this polyphenolic acid promotes the apoptosis of synovial cells. However, it should be also mentioned that the extract composition strongly depends on the extraction method, which was different in our research and research conducted by Huang, Syu and Lin [9]. Having in mind that EXT can directly influence *T. molitor* haemocytes and current literature data about the pro-apoptotic action of different EXT components, our subsequent study will focus on a detailed description of morphological and functional changes in the haemocytes after the application of tested extract and its main GAs. 

The second important part of the presented research was evaluating an impact of EXT on the essential activity of *T. molitor* immune mechanisms and after the additional immunisation of the tested beetles. Interestingly, reactions of the immune system of beetles with and without activated immune systems sometimes were not similar.

In the case of basic immune mechanisms, such as phenoloxidase activity and lysozyme-like antimicrobial activity, EXT applications significantly decreased their activity. The EXT action may be related to the cytotoxic effect of EXT on haemocytes (this article) and fat body [27], tissues strongly associated with regulation of phenoloxidase and lysozyme activity in insects [14,28]. Previous research by Spochacz et al. [27] demonstrated that changes in the fat body morphology appeared after the application of EXT at a concentration of 0.1%. Generally, the obtained results were similar in beetles with and without an activated immune system. However, it should be highlighted that lysozyme-like activity did not change in non-immunized beetles. This reaction of *T. molitor* lysozyme-like activity is partially consistent with research on tobacco hornworm *Manduca sexta*. Garvey et al. [29] presented that insects fed with recombined *Solanum lycopersicum*, characterised by higher toxicity than the wild type, reduced insect growth and development but did not reduce immune parameters, including PO activity. Moreover, the results obtained in the case of lysozyme-like activity of haemolymph collected from non-immunized beetles might be associated with the EXT influence on the expression level of immune-related genes in beetles without an activated immune system (this research).

The expression level of selected-immune genes significantly changed after the injection of EXT. However, the EXT effect depended on the tested beetles’ physiological state. In individuals treated only with physiological saline or solution of physiological saline and EXT, the upregulation of genes encoding tenecin 3 and Toll receptor was observed. There exists a possibility that increasing the expression level of gene encoding Toll receptors may be related to damage-associated molecular patterns (DAMPs) released from necrotic and apoptotic cells [30,31]. The presence of DAMPs triggers activation, among other things, the Toll pathway [30,31,32]. However, the immunostimulatory effect was observed only after the application of 0.01% EXT. For this reason, the action of a low dose of EXT may be connected with the hormesis process—a biphasic dose response, whereby exposure to low doses of a stressor can stimulate biological processes [33]. This phenomenon has also been reported in pest insects exposed to low doses of a pesticide [34,35] and plant secondary metabolites, including essential oils and alkaloids [36,37].

On the other hand, beetles with the activated immune system and treated with EXT were characterised by the lower expression levels of genes encoding cecropin and the Toll receptor, which might be associated with the general inhibition of immune mechanisms by EXT. There are three explanations for the reported results. First, similar to PO and lysozyme-like activity, this immunostatic effect can be associated with the cytotoxic effect of EXT on the haemocytes and fat body of *T. molitor* [27]. However, in non-immunized beetles, the downregulation of immune-related genes was not observed. The second possible explanation can be related to the fat body functions. Fat body tissue is one of the primary regulators of energy metabolism, nutrient storage, immune system activity, and detoxification [28,30]. We assume the decreased immune parameters in the bacterial-challenged beetles may result from the trade-off between the detoxification process and immune system activity. Interestingly, Dubovskiy et al. [38] obtained similar results. In that research, despite enhancing some immune mechanisms with a sub-lethal dose of nickel, *Galleria mellonella* larvae possessed lower resistance against pathogens [38]. Interestingly, research conducted by Shin et al. [39] and Zhao et al. [40] showed that solanine, one of the main components of EXT, elicited anti-inflammatory effects in murine macrophages activated by LPS. This immunostatic action of solanine was connected to the inhibition of nuclear translocation of nuclear factor-κB (NF-κB) and the suppression of activation of extracellular regulated protein kinases (ERK). These results suggest that EXT may directly regulate immune-related pathways, which is the third possible mode of action of the tested extract. For these reasons, further research is needed to explain this phenomenon fully. 

## 4. Conclusions

In summary, despite the well-documented influence of EXT on insect growth and metabolism, the present results are the first evidence of an immunomodulatory action of EXT in insects. Moreover, the application of EXT stimulates the apoptosis of *T. molitor* haemocytes, which may suggest impairment of their activity. The most important is that the more significant inhibition of immune system activity was observed in the case of immunised beetles, which may indicate the validity of the use of low-dose EXT as support for insecticides and alternative methods of biological control of pests. On the other hand, in the case of non-immunized beetles, EXT enhances the expression level of immune-related genes, which may induce a hormetic response. Based on this knowledge, future steps in this issue should be directed to the evaluation of EXT on cellular response and determination of the possibility of development in insects’ resistance against EXT and GAs. 

## 5. Materials and Methods

### 5.1. Insects

As a model organism in the research, adult individuals of *T. molitor* were used. Beetles were kept under stable conditions (28 °C, dark, 50–60% humidity) in the MIR 154-PE incubator (PHCbi, Singapore, Republic of Singapore). Purchased larvae were kept in plastic boxes with oatflakes and fresh apples (density—500 larvae per box) until they reach the pupal stage. The breeding protocol is based on the research conducted by El-Shazely et al. [41] and Urbański et al. [19]. To exclude the influence of oogenesis and senescence on *Tenebrio* immune system activity, in all experiments, only 7–8-day-old adult males were used (mean elytra size—9.67 ± 0.36 mm). After the *S. nigrum* fruit extract (EXT) application, beetles were kept in sterile square dishes with compartments (Sterilin™ 100 mm, Thermo-Fisher Scientific, Waltham, MA, USA).

### 5.2. Solanum Nigrum Fruit Extract

The extract from *S. nigrum* unripe berries was prepared at the laboratory of Professor Sabino A. Bufo, Potenza, Italy. The voucher specimens were deposited at the Herbarium Lucanum (HLUC, Potenza, Italy) with the ID Code: 2320. The extraction is based on the method previously described by Cataldi et al. [42]. Using a laboratory mill, the berry samples were lyophilized and ground to a fine powder. Next, samples (1.5 g) were placed in 20 mL of 1% acetic acid aqueous solution. The suspension was stirred for 2 h and then centrifuged at 5000× *g* for 30 min. The obtained pellet was suspended in 5 mL of 1% acetic acid, shaken, and centrifuged. The two supernatants were subsequently mixed. The extract was filtered through a single-use 0.22 µm nylon filter (Whatman, Maidstone, UK) and injected into the LC/MS system. The chemical analysis was conducted at the Department of Sciences, the University of Basilicata. The main components of the EXT are two GAs: solasonine and solamargine [7], The full characteristics of used EXT was previously published by Chowański et al. [7]. The EXT at concentrations of 0.01% or 0.1% were diluted in the physiological saline solution (274 mM NaCl, 19 mM KCl, 9 mM CaCl_2_). The used concentrations are based on recent literature data [6,27].

### 5.3. Injection and Tissue Collection

Before the injection of EXT or tissue collections, beetles were anaesthetised with endogenous CO_2_ by immersion in a beaker of clean water. Two microliters of physiological saline (Control) or EXT solution were injected with a Hamilton microliter syringe (Hamilton Company, Reno, NV, USA) under the coxa of the third pair of legs. Twenty-four hours after injection, haemolymph was collected by cutting the tibia of the first pair of legs. This method allowed us to obtain samples of clear haemolymph, without any contaminations. The fat body was dissected, 24 h after injection, using the Zeiss Stemi 508 microscope (Zeiss, Jena, Germany) under sterile conditions.

### 5.4. Total Haemocyte Count

The number of circulating haemocytes was determined according to Urbański et al. [43]. Twenty-four hours after injection of EXT, the hemolymph sample (2 µL) was gently mixed with the 20 µL of the physiological saline solution and the anticoagulation buffer (4.5 mM citric acid and 9 mM sodium citrate; 5:1, *v*/*v*) and then placed on the Bürker chamber. The photos of 24 randomly selected squares were taken with a Zeiss Primostar microscope equipped with Axiocam 105 digital camera (Zeiss, Jena, Germany). To evaluate the number of haemocytes on the photos, cells were counted using ImageJ software (version 2., public domain, https://imagej.net/software/imagej2/ (accessed on 20 October 2022)). At least 13 individuals were used per treatment (Control—16, EXT 0.01%—14 and EXT 0.1%—13).

### 5.5. Apoptosis

The indicator of the cytotoxic effect of EXT on the *T. molitor* haemocytes was the number of apoptotic cells estimated 24 h after injection of tested extract at concentrations of 0.01% and 0.1%. The number of apoptotic cells was determined in both treated and control samples based on the method described by Czarniewska et al. [44]. To detect active caspases in haemocytes, the sulforhodamine derivative of valyl alanyl aspartic acid fluoromethyl ketone, a potent inhibitor of caspase activity (SR-VAD-FMK, Enzo Life Sciences, Inc., New York, NY, USA), was used. In the beginning, the haemolymph sample (3 µL) was gently mixed with the physiological saline and anticoagulation buffer (5:1, *v*/*v*) and incubated on the microscopic slide coated with poly-L-lysine (Sigma Aldrich, Saint Louis, MO, USA) for 30 min in the dark. Then, the samples were washed 3-times with physiological saline. Next, the solution of SR-VAD-FMK and physiological saline (1:150 *v*/*v*) was added to the samples and incubated for 30 min in the dark. After this incubation, the residues were removed, and the samples were washed 3 times with a 10% wash buffer included in the SR-VAD-FMK kit. Next, samples were fixed with 4% paraformaldehyde (Sigma Aldrich, Saint Louis, MO, USA) for 15 min. Then, the specimens were washed 3 times with physiological saline, and the nucleic acid was stained with DAPI (Invitrogen, Thermo-Fisher Scientific, Waltham, MA, USA) for 7 min. Additionally, the direct effects of EXT on haemocytes were tested. In this case, samples were collected from non-injected individuals, and haemocytes had a contact with EXT only on microscopic slide, during 30 min incubation. For this treatment, haemocytes were incubated with the physiological saline solution with anticoagulation buffer (Control) or physiological saline with anticoagulation buffer and EXT at concentrations of 0.01% or 0.1%. The specimens were analysed using Nikon Eclipse TE 2000-U fluorescence microscope equipped with a Nikon DS-1QM camera (Nikon, Tokyo, Japan). Five images for each sample were taken. Photos were analysed using ImageJ (version 2, public domain, https://imagej.net/software/imagej2/ (accessed on 20 October 2022)) software, and the results were expressed as the percentage of apoptotic cells of the total number of haemocytes on the images. At least 5 individuals (370 haemocytes) were used per treatment (Control direct—6; EXT 0.01% direct—5; 0.1% direct—6; Control injection—6; EXT 0.01% injection—6; EXT 0.1% injection—6).

### 5.6. Phenoloxidase Activity

The activity of PO was determined based on the colorimetric method described by Sorrentino et al. [45] and Urbański et al. [46]. The EXT influence on PO activity was tested 24 h after its injection at 0.01% and 0.1% concentrations. Additionally, the PO activity was also measured after an additional injection of 2 µL 10% suspension of physiological saline and attenuated *Staphylococcus aureus* (Sigma Aldrich, Saint Louis, MO, USA) solution 2 h after application of EXT. This additional research variant may test the effect of EXT on the immunised beetles. A haemolymph sample (1 µL) was added to the Whatman No 52 filter paper (Whatman, Maidstone, UK) soaked with DL-DOPA (2 mg/1 mL, Sigma Aldrich, Saint Louis, MO, USA) in a 10 mM phosphate buffer to analyse the PO activity. Samples were incubated for 30 min at room temperature in the dark. Then, the samples were air-dried in the dark and scanned with SHARP AR 153 EN (600 dpi, 8 bits, grayscale, SHARP, Sakai, Japan), and analysed with ImageJ (version 2, public domain, https://imagej.net/software/imagej2/ (accessed on 20 October 2022)) software. Due to fact that lymph has not been separated from the cells, the recorded values also include the cellular PO activity. Results are expressed as percentage changes compared to control to better visualise the data. At least 10 individuals were used per treatment (Control—14; EXT 0.01%—14; EXT 0.1%—15; Control infected—10; EXT 0.01% infected—11; EXT 0.1% infected—11). 

### 5.7. The Lysozyme-like Activity of T. molitor Haemolymph

The method of determination of the lysozyme-like antimicrobial activity of haemolymph from *T. molitor* is based on the method described by Arce et al. [47] and Urbański et al. [48]. The tested individuals were injected with 2 µL of physiological saline solution and EXT at concentrations of 0.01% or 0.1%. The lysozyme-like activity was tested 24 h after applying the tested extract. Similarly, to previous bioassays, beetles with an activated immune system were tested. To activate the *T. molitor* immune system, two hours after physiological saline or EXT application, beetles were injected with 2 µL of a 10% physiological saline solution and attenuated *S. aureus* (Sigma Aldrich, Saint Louis, MO, USA). Twenty-four hours after the EXT injection, 2 µL of haemolymph samples were collected and transferred to 90 µL of ice-cold physiological saline and *Micrococcus luteus* solution (3 mg/10 mL; Sigma Aldrich, Saint Louis, MO, USA). Next, the samples were mixed and incubated at 37 °C for 30 min using a Thermomixer (Eppendorf, Hamburg, Germany). After incubation, samples were chilled on ice to inhibit the enzymatic reaction. The absorbance of the sample was checked (λ = 600 nm) using a BioSpectrometer kinetic (Eppendorf, Hamburg, Germany). The sample absorbance was compared to the absorbance of a physiological saline and *M. luteus* solution (blank; 0). As a positive control, the antimicrobial activity of a physiological saline and lysozyme solution (Sigma Aldrich, Saint Louis, MO, USA) was tested (0.1 mg/mL). At least 15 individuals were used in each treatment, and three independent replications were conducted (Control—15; EXT 0.01%—15; EXT 0.1%—15; Control infected—17; EXT 0.01% infected—21; EXT 0.1% infected—22).

### 5.8. Quantitative Analysis of Expression Level of Immune-Related Genes

To evaluate the changes in the expression level of immune-related genes, the Reverse Transcription quantitative PCR (RT-qPCR) according to the method described by Urbański et al. was used [48]. The expression level of the selected genes was determined in the primary immune-related tissues (pooled fat body and haemocytes), collected 24 h after application of EXT at concentrations of 0.01% and 0.1%. Moreover, quantitative changes in immune-related gene expression were also analysed after activation of the *T. molitor* immune system. For this purpose, two hours after injection of physiological saline or EXT, beetles were injected with 2 µL of a 10% physiological saline solution and attenuated by *S. aureus* (Sigma Aldrich, Saint Louis, MO, USA). The fat body and haemocytes were collected under sterile conditions using the Zeiss Stemi 508 microscope (Zeiss, Jena, Germany). The collected samples were placed in the RNA lysis buffer (Zymo Research, Irvine, CA, USA). Then, the samples were homogenised using electric pellets, frozen in liquid nitrogen and stored at −80 °C. Fat body and haemocytes from five individuals were collected for one biological replication. To isolate RNA, the Zymo MiniPrep kit (Zymo Research, Irvine, CA, USA) was used according to the manufacturer’s protocol. Next, the samples were treated with a Turbo DNase kit (Thermo-Fisher Scientific, Waltham, MA, USA). The quality of the samples and the RNA concentration were estimated using the H1 Hybrid Multi-Mode Microplate Reader (BioTek, Winooski, VT, USA). To transcript, an equal concentration of RNA (150 ng) to cDNA, RevertAid Kit (Thermo-Fisher Scientific, Waltham, MA, USA) was used. Additionally, to exclude external DNA contamination, a “no RT” experiment was performed (Appendix A).

The expression level of genes encoding two antimicrobial peptides: cecropin and tenecin 3, and the gene for Toll receptors, were analysed. The genes’ selection is based on the literature data [49,50]. Primers used in the experiment were synthesised by IBB Warsaw, based on the sequence of primers published previously by Jacobs et al. [49] (Appendix A). The RT-qPCR analyses were performed using SYBR Green Master mix (Thermo-Fisher Scientific, Waltham, MA, USA) according to the manufacturer’s protocol. Runs were performed using Corbett Research RG-6000 Real-Time PCR Thermocycler (Qiagen, Hilden, Germany). The relative expression of selected genes was normalised based on the expression level of the gene encoding *T. molitor* ribosomal protein L13a (TmRpL13a). Moreover, to confirm our results, the amplicons were sequenced by the Molecular Biology Techniques Laboratory (Faculty of Biology, Adam Mickiewicz University in Poznań) and compared with data available in a public database (NCBI, https://www.ncbi.nlm.nih.gov (accessed on 20 October 2022)). Three biological replicates were used for each treatment, and three technical replicates were performed. The relative expression was calculated using the 2^−ΔΔCt^ method [51].

### 5.9. Statistics

The obtained data using GraphPad Prism software were analysed (Department of Animal Physiology and Developmental Biology licence, version 9.0.0 for Windows, GraphPad Software, San Diego, CA, USA, www.graphpad.com (accessed on 20 October 2022)). The normality of the distribution was determined using the Shapiro-Wilk test. The Brown-Forsythe test and the Levene test were used to check the homogeneity of variance. Normally distributed data were analysed with Student’s *t*-test with Welch’s correction. Data with a non-normal distribution were analysed using the Mann-Whitney *U* test.

## Figures and Tables

**Figure 1 toxins-15-00068-f001:**
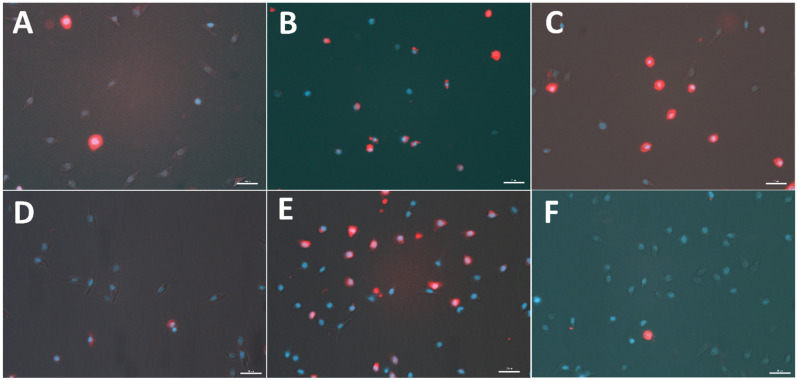
Representative micrographs of the cytotoxic activity of *S. nigrum* fruit extract (EXT) on haemocytes of *T. molitor*. (**A**–**C**)—haemocytes were incubated on the microscopic slides with the physiological saline solution (**A**) or physiological saline and EXT at concentrations of 0.01% (**B**) and 0.1% (**C**). (**D**–**F**)—haemocytes were collected 24 h after injection of physiological saline solution (**D**) or physiological saline and EXT at concentrations 0.01% (**E**) and 0.1% (**F**). Red—apoptotic haemocytes; blue—nucleic acids, scale bar—20 µm.

**Figure 2 toxins-15-00068-f002:**
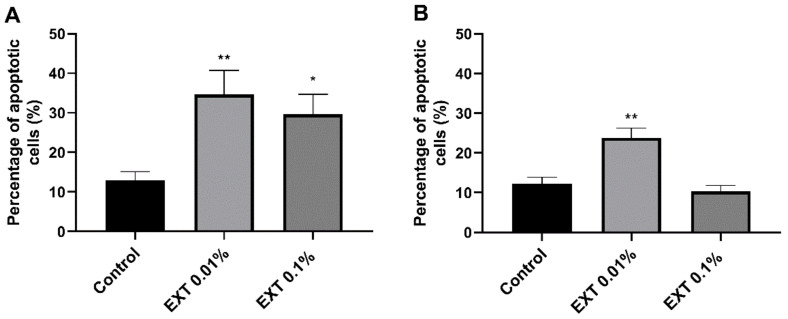
Percentage changes of apoptotic haemocytes after application of *S. nigrum* fruit extract (EXT) on haemocytes of *T. molitor*. (**A**)—haemocytes were incubated on the microscopic slides with the physiological saline solution or physiological saline and EXT at concentrations of 0.01% and 0.1%. (**B**)—haemocytes were collected 24 h after injection of physiological saline and EXT at concentrations of 0.01% and 0.1%. Values are shown as mean ± SEM. Asterisks—statistically significant changes compared to Control (* *p* ≤ 0.05; ** *p* ≤ 0.01).

**Figure 3 toxins-15-00068-f003:**
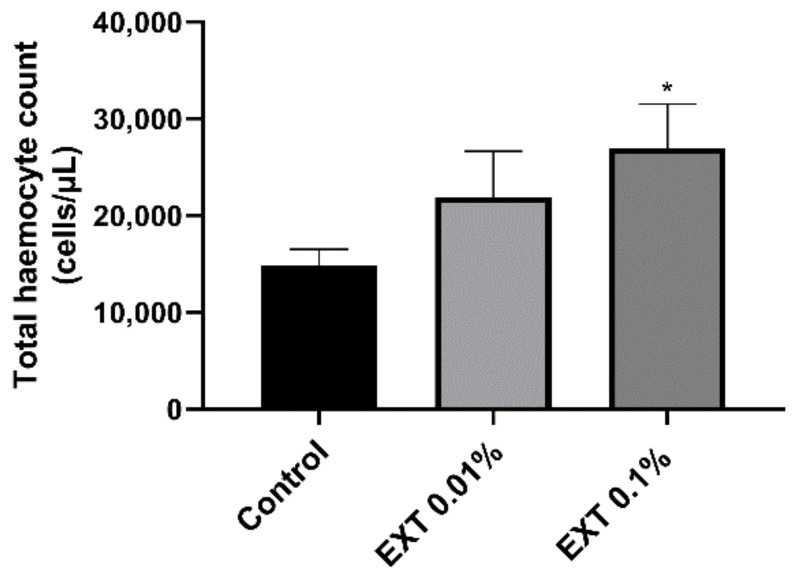
Total haemocyte count 24 h after injection of physiological saline solution (Control) or physiological saline and *S. nigrum* fruit extract (EXT) at concentrations of 0.01% and 0.1%. Values are shown as mean ± SEM. Asterisks—statistically significant changes compared to Control (* *p* ≤ 0.05).

**Figure 4 toxins-15-00068-f004:**
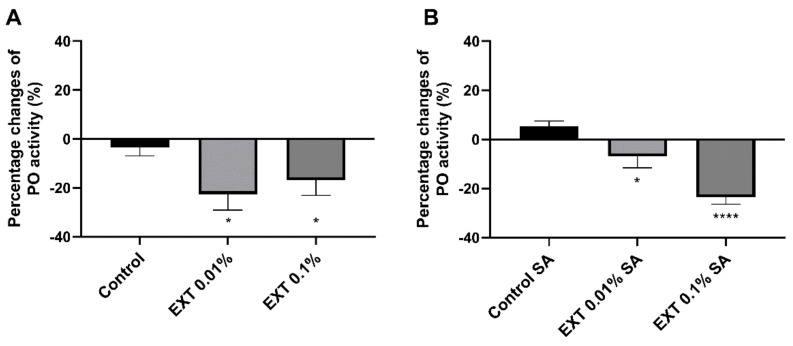
Percentage changes of phenoloxidase activity of *T. molitor* haemolymph 24 h after injection of physiological saline (Control) of *S. nigrum* fruit extract (EXT) at concentrations of 0.01% and 0.1%. (**A**)—haemolymph collected from non-immunized beetles; (**B**)—haemolymph collected after additional activation of *T. molitor* immune system with the suspension of physiological saline and attenuated *S. aureus* (SA). Values are shown as mean ± SEM. Asterisks—statistically significant changes compared to Control (* *p* ≤ 0.05, **** *p* ≤ 0.0001).

**Figure 5 toxins-15-00068-f005:**
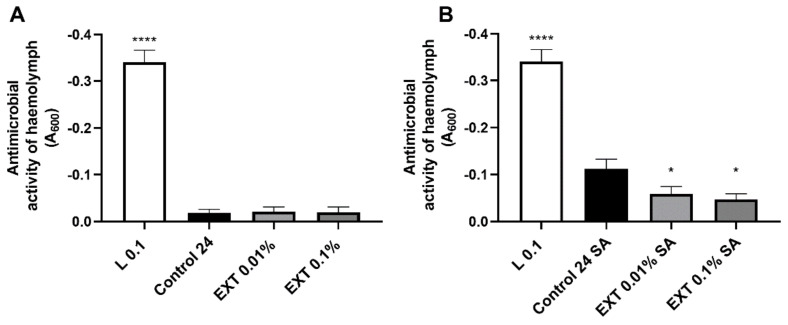
Changes in the lysozyme-like antimicrobial activity of *T. molitor* haemolymph after application of physiological saline (Control) or *S. nigrum* extract (EXT) at concentrations of 0.01 and 0.1%. (**A**)—haemolymph was collected from non-immunized beetles; (**B**)—haemolymph was collected from individuals additionally injected with a suspension of physiological saline and attenuated *S. aureus* (SA). L 0.1—positive control, solution of physiological saline and lysozyme at a concentration of 0.1 mg/mL. For better visualisation of the observed differences, the y-axis was reversed. Asterisks indicate statistically significant differences compared to the control individuals; * *p* ≤ 0.05; **** *p* ≤ 0.0001, values are given as means ± SEM.

**Figure 6 toxins-15-00068-f006:**
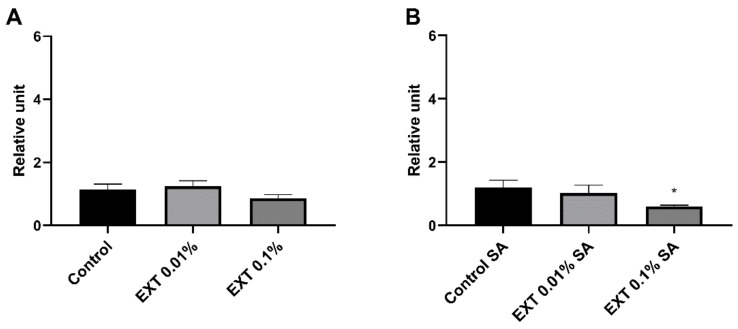
Reverse quantitative transcription PCR (RT-qPCR) analysis of expression level of gene encoding cecropin, one of the antimicrobial peptides of *T. molitor*, after application of physiological saline (Control) or *S. nigrum* extract (EXT) at concentrations of 0.01 and 0.1%. (**A**)—haemolymph was collected from beetles without an activated immune system; (**B**)—haemolymph was collected from individuals injected with a suspension of physiological saline and attenuated *S. aureus* (SA). Asterisks indicate statistically significant differences compared to the control individuals; * *p* ≤ 0.05; values are given as means ± SEM.

**Figure 7 toxins-15-00068-f007:**
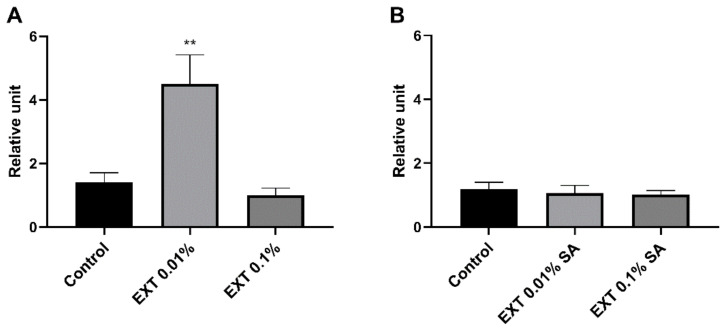
Reverse quantitative transcription PCR (RT-qPCR) analysis of expression level of gene encoding tenecin 3, one of the antimicrobial peptides of *T. molitor*, after application of physiological saline (Control) or *S. nigrum* extract (EXT) at concentrations of 0.01 and 0.1%. (**A**)—haemolymph was collected from beetles without an activated immune system; (**B**)—haemolymph was collected from individuals additionally injected with a suspension of physiological saline and attenuated *S. aureus* (SA). Asterisks indicate statistically significant differences compared to the control individuals; ** *p* ≤ 0.01; values are given as the means ± SEM.

**Figure 8 toxins-15-00068-f008:**
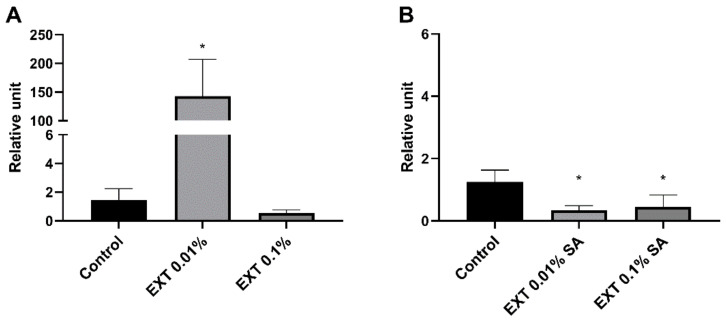
Reverse quantitative transcription PCR (RT-qPCR) analysis of expression level of gene encoding Toll receptor, one of the main parts of the Toll pathway, after application of physiological saline (Control) or *S. nigrum* extract (EXT) at concentrations of 0.01 and 0.1%. (**A**)—haemolymph was collected from beetles without an activated immune system; (**B**)—haemolymph was collected from individuals additionally injected with a suspension of physiological saline and attenuated *S. aureus* (SA). Asterisks indicate statistically significant differences compared to the control individuals; * *p* ≤ 0.05; values are given as means ± SEM.

## Data Availability

The datasets used during the current study are available from the corresponding author on reasonable request.

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
