# Peer review of "Solanum nigrum Fruit Extract Modulates Immune System Activity of Mealworm Beetle, Tenebrio molitor L."

_toxins, 2023, doi:10.3390/toxins15010068_

Round 1

Reviewer 1 Report

This manuscript is well organized and clearly written. However, it should be revised before it published. Please check my  comments and suggestions below. 

Introduction

Line 32-34: Not only the environmental change, also due to the pesticide resistant issue. So please include pesticide resistance issue with suitable reference.

Line 36-37: This sentence is not clear, please rephrase it.

Line 38-39: Solanum nigrum (Solanaceae Juss.) is a plant commonly distributed in Europe, not only in Europe, It is common in Asia and other parts of world. Please rephrase this line.

Results:

Line 67: Fig 1 and 2; Please write Figure 1 and 2. And correct this in whole manuscript as; Fig 3 to Figure 3 and so on.

Figures: Please check the fonts and size in all figure. Make uniform.

In Figure 4-8, what is the meaning of SA? Please mention in the foot note.

Conclusion: Please conclude your results more broadly and corelate with your objectives.

Materials and Methods

Insects: Please mention the insect rearing conditions more clearly. Eg; Photo period, food types etc.

Solanum nigrum fruit extract: From where you get this plants? Please mention the sources

Author Response

Dear Reviewer,

Thank you for delivering us the review of our manuscript: "Solanum nigrum fruit extract modulates immune system activity of mealworm beetle, Tenebrio molitor L.". Detailed response you can find below. 

This manuscript is well organized and clearly written. However, it should be revised before it published. Please check my comments and suggestions below. 

Introduction

Line 32-34: Not only the environmental change, also due to the pesticide resistant issue. So please include pesticide resistance issue with suitable reference.

According to Reviewer suggestion, the sentence was corrected.

Line 36-37: This sentence is not clear, please rephrase it.

Done

Line 38-39: Solanum nigrum (Solanaceae Juss.) is a plant commonly distributed in Europe, not only in Europe, It is common in Asia and other parts of world. Please rephrase this line.

Thank you very much for your comments. The sentence was revised.  

Results:

Line 67: Fig 1 and 2; Please write Figure 1 and 2. And correct this in whole manuscript as; Fig 3 to Figure 3 and so on.

Done

 Figures: Please check the fonts and size in all figure. Make uniform.

Thank you for the comment. Figures were revised.

In Figure 4-8, what is the meaning of SA? Please mention in the foot note.

Done. SA it is abbreviation of S. aureus

Conclusion: Please conclude your results more broadly and corelate with your objectives.

 Conclusion was connected with two main research goals.

Materials and Methods

Insects: Please mention the insect rearing conditions more clearly. Eg; Photo period, food types etc.

According to Reviewer suggestion, we modified description of rearing conditions

Solanum nigrum fruit extract: From where you get this plants? Please mention the sources

The information about EXT source was added to the Materials and Methods

Reviewer 2 Report

The manuscript titled “Solanum nigrum fruit extract modulates immune system activity of mealworm beetle, Tenebrio molitor L.” contains some important information. The study reported the first evidence of an immunomodulatory action of the extract of S. nigrum against T. molitor. The authors observed the significant inhibition of immune system activity in the case of immunised beetles, which may indicate the validity of the use of low-dose extracts as support for insecticides and alternative methods of biological control of pests.

However, there are some issues with the manuscript which need to be fixed.

The introduction is specific and focused on. However, the objectives of the study are missing. The authors should make the objectives clear, robust and concise.

Materials and methods are well explained. However, some of the protocols need to be substantiated with suitable references.  

In the materials and methods, the authors used 7–8-day-old adult males but did not mention their sizes. The authors should use the males of equal sizes.

Results are quite interesting, well-written and explained. However, the authors made use of present and past tenses in this section.  

Discussion is well written and is a reasonable analysis of the results. In some cases the authors made speculations. The authors are suggested to substantiate some of the statements in this section.

The conclusion needs revision and the authors should give some recommendation on the basis of their findings.

References section is good. However, there were inconsistencies in some of the references.

The language of the manuscript is good; however, there are some grammatical and syntactical errors throughout the manuscript which need to be fixed.

Author Response

Dear Reviewer,

Thank you for delivering us the review of our manuscript: "Solanum nigrum fruit extract modulates immune system activity of mealworm beetle, Tenebrio molitor L.". Detailed response can find below. 

The manuscript titled “Solanum nigrum fruit extract modulates immune system activity of mealworm beetle, Tenebrio molitor L.” contains some important information. The study reported the first evidence of an immunomodulatory action of the extract of S. nigrum against T. molitor. The authors observed the significant inhibition of immune system activity in the case of immunised beetles, which may indicate the validity of the use of low-dose extracts as support for insecticides and alternative methods of biological control of pests.

However, there are some issues with the manuscript which need to be fixed.

The introduction is specific and focused on. However, the objectives of the study are missing. The authors should make the objectives clear, robust and concise.

Thank you very much for your comments. The research objectives were present in previous version of the manuscript (lines 59-64, previously - lines 50-55).

Materials and methods are well explained. However, some of the protocols need to be substantiated with suitable references.  

In the previous version of the manuscript, all experimental protocols were supported by references.

In the materials and methods, the authors used 7–8-day-old adult males but did not mention their sizes. The authors should use the males of equal sizes.

According to Reviewer suggestion Materials and Methods was revised. Beetles are breading under stable conditions, including density and food amount. Also, the differences between random adult beetles are very low (elytra size -9.67±0,36 mm, mean±SEM). All these informations were added to Materials and Methods.

Results are quite interesting, well-written and explained. However, the authors made use of present and past tenses in this section.

Results section once again was checked and revised.

Discussion is well written and is a reasonable analysis of the results. In some cases the authors made speculations. The authors are suggested to substantiate some of the statements in this section.

Thank you for the comment. Some of the references were added to the Discussion. Speculative form of some of the conclusions was associated with the lack of knowledge about influence of EXT on insect immune system. For this reason, further research is needed to fully understand of EXT mode of action on insect immune cells.

The conclusion needs revision and the authors should give some recommendation on the basis of their findings.

The conclusion was revised.

References section is good. However, there were inconsistencies in some of the references.

Done

The language of the manuscript is good; however, there are some grammatical and syntactical errors throughout the manuscript which need to be fixed.

According to Reviewer suggestion the language was once again checked.

Reviewer 3 Report

The manuscript indicates potential impairments in the T. molitor immune system after the application of EXT. The manuscript shed light on the possible mechanisms of action of bioinsecticides and, therefore, has a prominent contribution to the area. There are a few points that need clarification before manuscript publication. Thus, my recommendation is a minor review.

L 181: Are they lethal or sublethal concentrations? Please indicate to be clear why they were used.

L 283-285: What was the volume of solution (control and EXT) applied per insect?

L 285-286: Was the haemolymph collected right after the application or 24 h? It is unclear.

L 297: This mention leads to imprecision about the number of insects used. Please provide the number of insects used in the analysis here and after (lines 321, 335, 354-355). The statistical analysis is unbalanced if the number of replicates per treatment is unequal.  

Author Response

Dear Reviewer,

Thank you for delivering us the review of our manuscript: "Solanum nigrum fruit extract modulates immune system activity of mealworm beetle, Tenebrio molitor L.". Detailed response you can find below.

The manuscript indicates potential impairments in the T. molitor immune system after the application of EXT. The manuscript shed light on the possible mechanisms of action of bioinsecticides and, therefore, has a prominent contribution to the area. There are a few points that need clarification before manuscript publication. Thus, my recommendation is a minor review.

L 181: Are they lethal or sublethal concentrations? Please indicate to be clear why they were used.

According to Spochacz et al. (2018) used concentrations are sub-lethal. The sentence was corrected.

L 283-285: What was the volume of solution (control and EXT) applied per insect?

Always we injected 2 µL. This information was added to the Materials and Methods.

L 285-286: Was the haemolymph collected right after the application or 24 h? It is unclear.

Always, samples were collected 24 hours after application. The sentence was modified.

L 297: This mention leads to imprecision about the number of insects used. Please provide the number of insects used in the analysis here and after (lines 321, 335, 354-355). The statistical analysis is unbalanced if the number of replicates per treatment is unequal. 

According to the Reviewer suggestion, detailed number of individuals was added in each section of Materials and Methods.

Reviewer 4 Report

Manuscript Number: toxins-2013237
Title: Solanum nigrum fruit extract modulates immune system activity of
mealworm beetle, Tenebrio molitor L.

Authors tested the effect of Solanum nigrum fruit extract on the cellular and humoral immune responses of T. molitor. There is a general need of knowledge on the bioinsecticides to counteract insect pests and T. monitor is a good model to perform preliminary lab analyses.

The final results are impressive and convincing. Therefore, I recommend the article for publication and suggest some modifications throughout the text.

Introduction

To improve this section, an adequate background of knowledge on haemocyte population and cellular and humoral immune responses in T molitor should be inserted (Line 49), adding the following references and similar:

Vigneron, A., Jehan, C., Rigaud, T., & Moret, Y. (2019). Immune defenses of a beneficial pest: the mealworm beetle, Tenebrio molitor. Frontiers in physiology, 10, 138.

Kojour, M. A. M., Baliarsingh, S., Am Jang, H., Yun, K., Park, K. B., Lee, J. E., ... & Jo, Y. H. (2022). Current knowledge of immune priming in invertebrates, emphasizing studies on Tenebrio molitor. Developmental & Comparative Immunology, 127, 104284.

Vommaro, M. L., Kurtz, J., & Giglio, A. (2021). Morphological characterisation of haemocytes in the mealworm beetle Tenebrio molitor (Coleoptera, Tenebrionidae). Insects, 12(5), 423.

as well as the chemical composition of extract to better argue in the discussion the observed biological effect in this model species

Huang, H. C., Syu, K. Y., & Lin, J. K. (2010). Chemical composition of Solanum nigrum linn extract and induction of autophagy by leaf water extract and its major flavonoids in AU565 breast cancer cells. Journal of agricultural and food chemistry, 58(15), 8699-8708.

Results

L 68 As stated in the M&M section, it is not clear why the authors written about in vitro treatment. Were the bacteria inoculated into the beetles or onto the slide? The text of experimental design should be revised in the M&M section and in related section of results.

L92-93 The concentration 0.01% is not significant perhaps because the number of samples used is very low

Fig2. L96-97 The reference to fig 1 should be revised as “(Fig.1D)”, “Fig.1E”… and so

Fig3 … after injection of physiological saline solution (control)….

Fig4 “control” and “SA” should be indicated in the text of caption, please revise all figs.

Fig. 8: the concentration reported in fig should be revised

Discussion

If the composition of the extract is considered, some results could be discussed in comparison with the biological activity of some components (see  Huang, H. C., Syu, K. Y., & Lin, J. K. (2010). Journal of agricultural and food chemistry, 58(15), 8699-8708). Polyphenols are the main component of extract and they are and antioxidant effects on cells see for ex. The follow references or similar

 Curti, V., Di Lorenzo, A., Dacrema, M., Xiao, J., Nabavi, S. M., & Daglia, M. (2017, October). In vitro polyphenol effects on apoptosis: An update of literature data. In Seminars in Cancer Biology (Vol. 46, pp. 119-131). Academic Press.

Halliwell, B. (2008). Are polyphenols antioxidants or pro-oxidants? What do we learn from cell culture and in vivo studies?. Archives of biochemistry and biophysics, 476(2), 107-112.

Singh, S., Kaur, I., & Kariyat, R. (2021). The multifunctional roles of polyphenols in plant-herbivore interactions. International Journal of Molecular Sciences, 22(3), 1442.

M&M

The entire section should be reviewed because concentrations of the solutions used are often not indicated. “Physiological saline” in a generic name, but it is required to specify at least in this session its composition in concentrations.

L283 what it means “endogenous CO2, please explain more

L284-286 What is the composition of the physiological saline solution, revise in all text. The amount injected and the amount of haemolymph collected from each specimens should be indicated

L301 The number of apoptotic cells was determined in both treated and control samples based on…

L307-308 The concentration of solution is not clear, please revise

L314-315 In vitro cell treatment appears to have been done. The sentence should be revised.

L321 The number of specimens used is very low

L325 the amount of extract injected should be indicated

L323-334  Although the method used has already been tested in previous papers, a protocol in which PO activity is measured spectrophotometrically as the variation of abs per microliter per min is more appropriate to evaluate the melanisation ability of specimens under treatment. Moreover, if the haemolymph has not been separated from the cells, the recorded values also include the cellular PO activity.

L349 usually Lysozyme-like activity is checked at 450 nm

Author Response

Dear Reviewer,

Thank you for delivering us the review of our manuscript: "Solanum nigrum fruit extract modulates immune system activity of mealworm beetle, Tenebrio molitor L.". Detailed response you can find below.

Authors tested the effect of Solanum nigrum fruit extract on the cellular and humoral immune responses of T. molitor. There is a general need of knowledge on the bioinsecticides to counteract insect pests and T. monitor is a good model to perform preliminary lab analyses.

The final results are impressive and convincing. Therefore, I recommend the article for publication and suggest some modifications throughout the text.

Thank you very much for your opinion

Introduction

To improve this section, an adequate background of knowledge on haemocyte population and cellular and humoral immune responses in T molitor should be inserted (Line 49), adding the following references and similar:

Vigneron, A., Jehan, C., Rigaud, T., & Moret, Y. (2019). Immune defenses of a beneficial pest: the mealworm beetle, Tenebrio molitor. Frontiers in physiology, 10, 138.

Kojour, M. A. M., Baliarsingh, S., Am Jang, H., Yun, K., Park, K. B., Lee, J. E., ... & Jo, Y. H. (2022). Current knowledge of immune priming in invertebrates, emphasizing studies on Tenebrio molitor. Developmental & Comparative Immunology, 127, 104284.

Vommaro, M. L., Kurtz, J., & Giglio, A. (2021). Morphological characterisation of haemocytes in the mealworm beetle Tenebrio molitor (Coleoptera, Tenebrionidae). Insects, 12(5), 423.

as well as the chemical composition of extract to better argue in the discussion the observed biological effect in this model species

Huang, H. C., Syu, K. Y., & Lin, J. K. (2010). Chemical composition of Solanum nigrum linn extract and induction of autophagy by leaf water extract and its major flavonoids in AU565 breast cancer cells. Journal of agricultural and food chemistry, 58(15), 8699-8708.

According to the Reviewer suggestion, Introduction was revised.

Results

L 68 As stated in the M&M section, it is not clear why the authors written about in vitro treatment. Were the bacteria inoculated into the beetles or onto the slide? The text of experimental design should be revised in the M&M section and in related section of results.

Thank you for the comment. In case of the apoptotic ratio, the research variant with bacteria were not performed. As we stated in the manuscript “This cytotoxic effect was observed after EXT injection and in “in vitro” treatment when the tested extract was added to the physiological saline during the incubation of haemocytes on microscopic slides.” In “in vitro” treatment, haemocytes were treated with EXT only during 30 min incubation on the microscopic slide. The sentence was slightly modified.

L92-93 The concentration 0.01% is not significant perhaps because the number of samples used is very low

Thank you very much for your comments. In case of THC value, the minimal number of individuals per treatment was 13 (control – 16, 0.01% EXT – 14 and 0.1% EXT – 13). Based on our experience and research performed by specialist in the field, we assume that number of individuals is appropriate (Ebrahimi et al., 2014; Giglio et al., 2008; Urbański et al., 2014; Urbański et al., 2017; Urbański et al., 2021a; Urbański et al., 2021b).

Fig2. L96-97 The reference to fig 1 should be revised as “(Fig.1D)”, “Fig.1E”… and so

The Figure caption was corrected.

Fig3 … after injection of physiological saline solution (control)….

Done

Fig4 “control” and “SA” should be indicated in the text of caption, please revise all figs.

Done

Fig. 8: the concentration reported in fig should be revised

The concentrations in Fig. 8 are correct.

Discussion

If the composition of the extract is considered, some results could be discussed in comparison with the biological activity of some components (see Huang, H. C., Syu, K. Y., & Lin, J. K. (2010). Journal of agricultural and food chemistry, 58(15), 8699-8708). Polyphenols are the main component of extract and they are and antioxidant effects on cells see for ex. The follow references or similar

 Curti, V., Di Lorenzo, A., Dacrema, M., Xiao, J., Nabavi, S. M., & Daglia, M. (2017, October). In vitro polyphenol effects on apoptosis: An update of literature data. In Seminars in Cancer Biology (Vol. 46, pp. 119-131). Academic Press.

Halliwell, B. (2008). Are polyphenols antioxidants or pro-oxidants? What do we learn from cell culture and in vivo studies?Archives of biochemistry and biophysics, 476(2), 107-112.

Singh, S., Kaur, I., & Kariyat, R. (2021). The multifunctional roles of polyphenols in plant-herbivore interactions. International Journal of Molecular Sciences, 22(3), 1442.

Thank you for your valuable comment. The Discussion was revised.

M&M

The entire section should be reviewed because concentrations of the solutions used are often not indicated. “Physiological saline” in a generic name, but it is required to specify at least in this session its composition in concentrations.

We agree with Reviewer for the importance of this information. For this reason, physiological saline composition was included in the first version of the manuscript. You can find this information in line 323 (previously 280).

L283 what it means “endogenous CO2, please explain more

According to Reviewer suggestion, we added additional explanation.

L284-286 What is the composition of the physiological saline solution, revise in all text. The amount injected and the amount of haemolymph collected from each specimens should be indicated.

Please see previous comment. We added missing information.

L301 The number of apoptotic cells was determined in both treated and control samples based on…

Thank you for suggestion, the sentence was revised.

L307-308 The concentration of solution is not clear, please revise

The concentration was added.

L314-315 In vitro cell treatment appears to have been done. The sentence should be revised.

The sentences were corrected, and this variant was better explained.

L321 The number of specimens used is very low

For each research variants, at least 5 individuals were used (Control direct – 6; EXT 0.01% direct – 5; 0.1% direct – 6; Control injection – 6; EXT 0.01% injection – 6; EXT 0.1% injection – 6). Based on our experience and research published by specialist in the field, we think that number of used individuals is sufficient. It is proven by existence of statistically significant differences between groups. Also, the EXT effect was strong and clearly visible. The examples of research on insect haemocytes in which similar number of individuals was used (Giglio et al., 2008; Giglio et al., 2015; Lubawy et al., 2019; Urbański et al., 2021a; Wan et al., 2020).

L325 the amount of extract injected should be indicated

This information was added to the section “5.3. Injection and tissue collection”.

L323-334 Although the method used has already been tested in previous papers, a protocol in which PO activity is measured spectrophotometrically as the variation of abs per microliter per min is more appropriate to evaluate the melanisation ability of specimens under treatment. Moreover, if the haemolymph has not been separated from the cells, the recorded values also include the cellular PO activity.

Thank you for the suggestion, this information was added to the Material and Methods.

L349 usually Lysozyme-like activity is checked at 450 nm

Our method based on the method previously published by Arce et al. (2013) and Urbański et al. (2022). The main indicator of lysozyme-like activity was measurement of bacterial suspension clearance after incubation with haemolymph. For this reason, wavelength 600 nm was used (classical wavelength for estimation of bacteria concentration).

REFERENCES

  Arce, A. N., Smiseth, P. T. and Rozen, D. E. (2013). Antimicrobial secretions and social immunity in larval burying beetles, Nicrophorus vespilloides. Animal Behaviour 86, 741-745.

  Chowański, S., Chudzińska, E., Lelario, F., Ventrella, E., Marciniak, P., Miądowicz-Kobielska, M., Spochacz, M., Szymczak, M., Scrano, L. and Bufo, S. A. (2018). Insecticidal properties of Solanum nigrum and Armoracia rusticana extracts on reproduction and development of Drosophila melanogaster. Ecotoxicol Environ Saf 162, 454-463.

  Ebrahimi, L., Niknam, G., Dunphy, G. B. and Toorchi, M. (2014). Effect of an entomopathogenic nematode, Steinernema carpocapsae on haemocyte profile and phenoloxidase activity of the Colorado potato beetle, Leptinotarsa decemlineata. Biocontrol Science and Technology 24, 1383-1393.

  Giglio, A., Battistella, S., Talarico, F. F., Brandmayr, T. Z. and Giulianini, P. G. (2008). Circulating hemocytes from larvae and adults of Carabus (Chaetocarabus) lefebvrei Dejean 1826 (Coleoptera, Carabidae): Cell types and their role in phagocytosis after in vivo artificial non-self-challenge. Micron 39, 552-558.

  Giglio, A., Brandmayr, P., Pasqua, T., Angelone, T., Battistella, S. and Giulianini, P. G. (2015). Immune challenges trigger cellular and humoral responses in adults of Pterostichus melas italicus (Coleoptera, Carabidae). Arthropod Structure & Development 44, 209-217.

  Lubawy, J., Urbański, A., Mrówczyńska, L., Matuszewska, E., Światły-Błaszkiewicz, A., Matysiak, J. and Rosiński, G. (2019). The influence of bee venom melittin on the functioning of the immune system and the contractile activity of the insect heart—A preliminary study. Toxins 11, 494.

  Spochacz, M., Chowański, S., Szymczak, M., Lelario, F., Bufo, S. A. and Adamski, Z. (2018). Sublethal effects of Solanum nigrum fruit extract and its pure glycoalkaloids on the physiology of Tenebrio molitor (Mealworm). Toxins 10, 504.

  Urbański, A., Czarniewska, E., Baraniak, E. and Rosinski, G. (2014). Developmental changes in cellular and humoral responses of the burying beetle Nicrophorus vespilloides (Coleoptera, Silphidae). Journal of Insect Physiology 60C, 98-103.

  Urbański, A., Czarniewska, E., Baraniak, E. and Rosiński, G. (2017). Impact of cold on the immune system of burying beetle, Nicrophorus vespilloides (Coleoptera: Silphidae). Insect Science 24, 443-454.

  Urbański, A., Johnston, P., Bittermann, E., Keshavarz, M., Paris, V., Walkowiak-Nowicka, K., Konopińska, N., Marciniak, P. and Rolff, J. (2022). Tachykinin-related peptides modulate immune-gene expression in the mealworm beetle Tenebrio molitor L. Sci Rep 12, 1-19.

  Urbański, A., Konopińska, N., Lubawy, J., Walkowiak-Nowicka, K., Marciniak, P. and Rolff, J. (2021a). A possible role of tachykinin-related peptide on an immune system activity of mealworm beetle, Tenebrio molitor L. Developmental & Comparative Immunology 120, 104065.

  Urbański, A., Walkowiak-Nowicka, K., Nowicki, G., Chowański, S. and Rosinski, G. (2021b). Effect of short-term desiccation, recovery time and CAPA-PVK neuropeptides on the immune system of the burying beetle Nicrophorus vespilloides. Front Physiol 12, 845.

  Wan, B., Yang, L., Zhang, J., Qiu, L., Fang, Q., Yao, H., Poirié, M., Gatti, J.-L. and Ye, G. (2020). The venom of the ectoparasitoid wasp Pachycrepoideus vindemiae (Hymenoptera: Pteromalidae) induces apoptosis of Drosophila melanogaster hemocytes. Insects 11, 363.

Round 2

Reviewer 4 Report

The responses were accurate and the revision satisfactory. I'd like to suggest the paper for publication.

Author Response

Dear Reviewer and Editor,

Thank you for your comment. Response to Editor Comments was submitted in the different section. 

Best regards

Arkadiusz Urbański